# Bisdemethoxycurcumin Reduces Methicillin-Resistant *Staphylococcus aureus* Expression of Virulence-Related Exoproteins and Inhibits the Biofilm Formation

**DOI:** 10.3390/toxins13110804

**Published:** 2021-11-15

**Authors:** Shu Wang, Ok-Hwa Kang, Dong-Yeul Kwon

**Affiliations:** Department of Oriental Pharmacy, College of Pharmacy and Wonkwang Oriental Medicines Research Institute, Wonkwang University, Iksan 54538, Jeonbuk, Korea; wshu1996@gmail.com

**Keywords:** MRSA, α-toxin, enterotoxin A, enterotoxin B, biofilm, bisdemethoxycurcumin

## Abstract

Methicillin-resistant *Staphylococcus aureus* (MRSA) is a major pathogen of nosocomial infection, which is resistant to most antibiotics. Presently, anti-virulence therapy and anti-biofilm therapy are considered to be promising alternatives. In the current work, we investigated the influence of bisdemethoxycurcumin (BDMC) on the virulence-related exoproteins and the biofilm formation using a reference strain and clinic isolated strains. Western blotting, quantitative RT-PCR, and tumor necrosis factor (TNF) release assay were performed to assess the efficacy of BDMC in reducing the expression of *Staphylococcus* enterotoxin-related exoproteins (enterotoxin A, enterotoxin B) and α-toxin in MRSA. The anti-biofilm activity of BDMC was evaluated through a biofilm inhibition assay. The study suggests that sub-inhibitory concentrations of BDMC significantly inhibited the expression of *sea*, *seb*, and *hla* at the mRNA level in MRSA. Moreover, the expression of virulence-related exoproteins was significantly decreased by down-regulating accessory gene regulator *agr*, and the inhibition of biofilms formation was demonstrated by BDMC at sub-inhibitory concentrations. Consequently, the study suggests that BDMC may be a potential natural antibacterial agent to release the pressure brought by antibiotic resistance.

## 1. Introduction

Methicillin-resistant *Staphylococcus aureus* (MRSA) is a group of Gram-positive bacteria causing infections in communities and medical institutions worldwide, with a high prevalence and severe drug resistance [1,2]. MRSA causes skin and tissue infections to patients, which, in severe cases, can cause endocarditis, sepsis and necrotizing pneumonia [3]. Unfortunately, drug-resistant strains have been reported through the use of vancomycin and linezolid, which are the current guideline-advised treatments for MRSA [4]. Hence, it is necessary to continuously develop novel therapies, including the use of antimicrobial adjuvants, anti-virulence, and anti-biofilm antibacterial [5].

*Staphylococcus aureus* (*S. aureus*) produces a variety of toxins during the growing process, including hemolysins, enterotoxins, leukocidin and toxic shock syndrome toxin [6]. α-Toxin (α-hemolysin) is an exocrine protein encoded by the *hla* gene, which is cytotoxic to a variety of host cells and is also a toxin that causes cell damage and death [7]. Moreover, several studies have confirmed α-toxin production correlated significantly with MRSA biofilm formation [8]. Biofilm is a community of microorganisms attached to biotic and abiotic surfaces [9], exhibiting drug resistance to broad-spectrum antibiotics. This biofilm is recalcitrant in efforts to be eradicated by anti-bacterial agents, further limiting the efficacy of currently available antibiotics [10]. Biofilm formation is regulated by the quorum-sensing (QS) system [11], and the *hld* gene is regarded as the most effective molecule in the Staphylococcal QS system [12]. The biofilm formation increases MRSA resistance to antibiotics by 10–10,000 times and is also the cause of many chronic MRSA infections [13]. Therefore, ideal antibacterial agents could inhibit biofilm formation.

*Staphylococcal* enterotoxins (SEs) are a highly toxic family of proteins, which could cause food poisoning and even toxic shock syndrome (TSS). The first enterotoxin SEA was isolated in 1959, and so far, 25 SEs (SEA-SElZ) have been described [14]. These peptides are classified as pyrogenic toxin superantigens (SAg), which can mobilize a large number of T cells [15], resulting in the release of a large number of pro-inflammatory cytokines (such as IL-6 and TNF-α), leading to toxic shock syndrome (TSS) [16]. Characterized by high fever, rash and hypertension, it can rapidly progress to multiple organ failure and death, among which, *Staphylococcal* enterotoxin B (SEB) is a typical enterotoxin. It was developed as a biological weapon in the 20th century because of its high toxicity and stability [17].

Turmeric, which originated in Asia, has been used as a spice and traditional medicine for centuries [18]. The main biologically active component of turmeric is curcumin, including the structurally related analogs, desmethoxycurcumin, and bisdemethoxycurcumin (BDMC), of which BDMC is a polyphenolic curcuminoid, with a variety of pharmacological activities, such as anti-inflammatory, antioxidant and anti-cancer effects [19,20]. However, currently, the antibacterial activity of BDMC is rarely studied. In our previous study, the antimicrobial activity of BDMC against MRSA has been proved with a MIC value of as low as 7.8 µg/mL, which was resistant to the reference strain ATCC 33591 and clinically isolated strains [21]. The antibacterial mechanism of MRSA was further studied in the current report. The inhibitory effect of sub-inhibitory concentrations of BDMC on the expression of α-toxin and enterotoxin protein as well as the regulatory genes *hla*, *agrA*, *sea* and *seb* [22] have been studied. Thus, the inhibitory effect of BDMC on biofilm has been studied in the present work. 

## 2. Results

### 2.1. BDMC Reduces the Expression of α-Toxin and Inhibits the Expression of agrA and hla in S. aureus

Western blot was examined to evaluate the level of expression of α-toxin treated or untreated with BDMC of sub-inhibitory concentrations. After treatment for 4 h, BDMC production of α-toxin was determined to be significantly suppressed. In the presence of 1/8 MIC of BDMC, the protein level of α-toxin was only detected at 11%. However, as the concentration of BMDC increases, protein levels no longer decrease significantly (Figure 1a). The transcriptional level of the α-toxin gene *hla* and *agrA* was quantified by quantitative RT-PCR (qRT-PCR). The concentration of BDMC significantly downregulated the expression of *agrA* and *hla* as low as 0.9 µg/mL (1/8 MIC) (Figure 1b,c). 

### 2.2. BDMC Inhibits Biofilm Formation and Downregulated the Expression of hld

A crystal violet biofilm assay has been performed to evaluate the inhibition of BDMC in biofilm formation at sub-inhibitory concentrations (1/8 MIC, 1/4 MIC, and 1/2 MIC). Dose-dependent BDMC inhibited biofilm formation significantly at sub-inhibitory concentrations. Specifically, at 1/2 MIC, BDMC inhibited biofilm formation of ATCC 33591, DPS-1, and DPS-2 by 78%, 82%, and 98%, respectively (Figure 2a). The results indicated that the production of α-toxin was positively correlated with the formation of MRSA biofilm. Additionally, the transcriptional level of the biofilm-related gene *hld* was quantified by qRT-PCR. BDMC at 3.9 µg/mL (1/2 MIC) significantly downregulated the expressions of *hld* (Figure 2b).

### 2.3. Expression of Staphylococcus Enterotoxin-Related Exoproteins in MRSA Treated with BDMC

Western blot was performed to evaluate the level of expression of *Staphylococcus enterotoxin*-related exoproteins (enterotoxin A and enterotoxin B) treated or untreated with BDMC. After exposure to treatment for 4 h, both exotoxins were inhibited by treating with sub-inhibitory concentrations of BDMC. The result showed that enterotoxin A and enterotoxin B were induced after being treated with 1/8 MIC of BDMC. In the presence of 1/4 MIC of BDMC, enterotoxin A was slightly inhibited, and enterotoxin B was significantly inhibited. Moreover, the production of enterotoxin B production in *S. aureus* ATCC 33591 was almost undetectable in the presence of 1/2 MIC (3.9 µg/mL) of BDMC (Figure 3a). 

### 2.4. BDMC Downregulated the Expression of sea and seb in MRSA

A qRT-PCR was performed to investigate the expression of the *Staphylococcus* enterotoxin genes (*sea* and *seb*) after treating with BDMC at sub-inhibitory concentrations. The expression of *sea* and *seb* was significantly down-regulated in *S. aureus* ATCC 33591 in a dose-dependent manner. The best inhibitory effect was shown after exposure to 1/2 MIC (3.9 µg/mL) of BDMC. Meanwhile, the transcriptional level of *sea* and *seb* were decreased by 1.7-fold and 5-fold, respectively (Figure 3b,c).

### 2.5. BDMC Reduces the Activity of IL-6 and TNF-α 

As mentioned above, enterotoxin A and enterotoxin B, as superantigens, were produced by *S. aureus*, which could trigger an immune response, leading to proinflammatory cytokines. Enzyme-linked immunosorbent assay (ELISA) was used to determine the proinflammatory cytokines TNF-α and IL-6 levels in enterotoxin-stimulated RAW 264.7 cells, which grew in the culture supernatants of *S. aureus* that was treated with sub-inhibitory concentrations of BDMC. The result indicated that the expression of IL-6 and TNF-α decreased in a dose-dependent manner. Moreover, the expression of TNF-α was significantly reduced (Figure 4b), and that the expression of IL-6 was slightly reduced (Figure 4a).

## 3. Discussion

With the widespread appearance of virulent and multidrug-resistant MRSA strains, the morbidity and mortality caused by *S. aureus* infection had increased [23]. On the one hand, *S. aureus* produces a significant quantity of secreted toxins, which is suspected to be a potential mechanism for *S. aureus* to establish a chronic infection [24]. In the current study, we investigated the inhibitory effect of BDMC on α-toxin, enterotoxine A and enterotoxin B of MRSA. The results indicated that sub-inhibitory concentrations of BDMC significantly inhibited the expression of *hla* at the mRNA level in MRSA and significantly reduced the α-toxin production of *S. aureus*. Moreover, BDMC reduced the expression of *Staphylococcal* accessory gene regulator *agrA*. Therefore, we speculated that BDMC inhibits α-toxin production by down-regulating the *agrA* gene. The results of qRT-PCR showed that sub-inhibitory concentrations of BDMC down-regulated the expression of enterotoxin regulatory genes *sea* and *seb*, which was consistent with the results of inhibition of enterotoxin proteins indicated by Western blotting. The dose-dependent decrease in the level of pro-inflammatory cytokines TNF-α and IL-6 in RAW 264.7 cells grown in *S. aureus* treated with sub-inhibitory concentrations of BDMC, which strengthened the evidence of the inhibitory effect of BDMC on enterotoxin. According to the above results, we speculated that reducing the expression level of enterotoxin protein by down-regulating related genes was one of the reasons why BDMC could inhibit enterotoxin. In addition, it has been reported that enterotoxins could bind to polyphenols [25], and BDMC is a polyphenolic compound with two phenolic hydroxyl groups. The hydrophobicity and hydrogen bonding property of the phenolic hydroxyl group are believed to be easily combined with proteins and this chemical structure could be the second reason of the enterotoxins inhibition. Since drugs that resist virulence factors are considered not to affect viability, which may not destroy beneficial bacteria flora, inhibiting virulence factors of pathogens may be a promising strategy against MRSA [4].

On the other hand, since all currently available antibiotics will inevitably develop resistance, the development of novel antibiotic replacement therapies is urgently required. Naturally sourced antibacterial agents are considered to have extremely low drug resistance due to their rich sources and diverse and complex structures [26]. Based on this, we additionally conducted a preliminary trial of the BDMC resistance test, and the results showed that BDMC had no apparent resistance after 10 consecutive passages (45 days), and the resistance effect was better than that of the traditional antibiotic linezolid in the control group. Moreover, the US Food and Drug Administration (FDA) has approved curcuminoids as “Generally Recognized as Safe” (GRAS) [27], and it is assumed in the literature report dedicated to the evaluation of the biological activity of curcumin that BDMC is more stable than the other two derivatives (curcumin and dimethoxycurcumin) [28]. Additionally, it has also been proved in our previous studies that the anti-MRSA ability of BDMC is also more potent than that of the other two derivatives [21]. These advantages increase the possibility of BDMC as a substitute for antibiotics. However, the low solubility of curcumin in aqueous media, poor bioavailability, and pharmacokinetic defects limit its application in clinical treatment. Given the limitations of turmeric, approaches, such as lipid or nanoparticle materials combined with curcumin to make a mixture, have been reported [29].

Additionally, the formation of biofilms is one of the reasons why MRSA is difficult to eradicate [30]. Biofilms could enhance resistance to antibiotics, and biofilm-related infections are usually chronic or recurrent. Therefore, in the current requirements for antibacterial substances, not only should bacteria be inhibited for growth, but there should also be the ability to inhibit biofilm formation [31]. Studies have shown that, in addition to inhibiting toxins, plant-derived polyphenols can also inhibit the biofilm formation of food-related pathogens [32]. Based on the above, we hypothesized that BDMC could inhibit MRSA biofilm. The result was assumed that BDMC had a potent inhibitory effect on the biofilm formation of the clinical strain and the reference strain ATCC 33591 at sub-inhibitory concentrations, and the effective concentration was as low as 0.9 µg/mL (1/8 MIC). Moreover, BDMC significantly down-regulated biofilm formation-related gene *hld*, which emphasized the inhibitory effect of BDMC on biofilm formation. In addition, it was reported that the production of toxins is one of the reasons leading to the production of biofilm. Therefore, we speculated that BDMC might also inhibit α-toxin production by the agr system, which would inhibit biofilm formation. Since biofilms are usually highly resistant to conventional antibiotics, the development of natural antibacterial substances with anti-biofilm activity may be of great significance [33].

In conclusion, our study first proposed the anti-toxin and anti-biofilm effects of BDMC, provided alternative or enhanced strategies for preventing and treating toxin-related infections, and provided a foundation for developing novel antibiofilm-specific antibiotics. Furthermore, our study revealed the potential of BDMC as a natural antibacterial agent, and a reference value was provided for the development of an antibiotic replacement, antibiotic adjuvant, or antibacterial lead compound. However, this study has certain limitations, such as the lack of a variety of tested strains and the lack of in vivo experiments. If the application of curcumin in clinical treatment is promoted, more in-depth in vivo experiments and the improvement of pharmacokinetics will be needed.

Furthermore, to solve the problem of MRSA infection, it is also critical to strengthen the solution to the occurrence of infection in communities. Reports have shown that the current spread of MRSA in the population is no longer caused by a large-scale nosocomial outbreak, and the proportion of community-related invasiveness as well as infections is higher [34].

## 4. Materials and Methods

### 4.1. Reagents

BDMC was purchased from Tokyo Chemical Industry Co.; Ltd. (Tokyo, Japan). Skim milk, Mueller–Hinton agar (MHA), and Mueller–Hinton broth (MHB) were obtained from Difco Laboratories (Baltimore, MD, USA). Anti-Staphylococcus enterotoxin A antibody, anti-Staphylococcus enterotoxin B antibody, and anti-Staphylococcus alpha-hemolysin antibody were acquired from Abcam (Cambridge, UK). Anti-rabbit IgG secondary antibody was acquired from Thermo Scientific Inc. (Waltham, MA, USA). Crystal violet was obtained from Sigma-Aldrich Co. (St. Louis, MO, USA). The sequences of primers used in qRT-PCR were listed in Table 1, which was acquired from Bioneer (Daejeon, Korea).

### 4.2. Bacterial Strains and Growth Conditions

In the report, one reference strain *S. aureus* ATCC 33591 (American Type Culture Collection, Manassas, VA, USA) was used as a reference strain and two clinical isolates of MRSA (DPS-1 and DPS-2) were isolated from patients at the Hospital of Wonkwang University was used. *S. aureus* was cultured in an incubator at 37 °C, and MHA or MHB was used as a solid medium or liquid medium, respectively.

### 4.3. Western Blot Assay

A Western blot assay was performed to evaluate the effect of BDMC on the expression of virulence-related exoproteins based on the previously reported investigation [21]. *S. aureus* strains (ATCC 33591) were cultured in MHB for 24 h and then treated for 4 h with sub-inhibitory concentrations of BDMC. Protein concentration was measured using a Bio-Rad protein assay reagent (Bio-Rad Laboratories, Hercules, CA, USA) after centrifugation at 3000× *g* for 10 min to harvest cell protein extracts. The supernatant was separated by SDS-PAGE and electroblotted onto Amersham HybondTM-P membranes (GE Healthcare, Piscataway, NJ, USA) that had been blocked with 5% skim milk and probed for 8 h at 4 °C with anti-Staphylococcus alpha-hemolysin antibody, anti-Staphylococcus enterotoxin A antibody, and anti-Staphylococcus enterotoxin B antibody (diluted 1:1000), and then reprobed by anti-rabbit IgG secondary antibody (diluted 1:1000) at room temperature for 2 h. The membranes were then supplemented with ECL^™^ Prime Western Blotting Detection reagent (GE Healthcare Life Sciences, Incheon, Korea). The bands were visualized using the ImageQuant LAS-4000 mini chemical luminescent imager (GE Healthcare Life Sciences).

### 4.4. Reverse Transcription and qRT-PCR

A qRT-PCR was carried out according to the methods previously described [35]. Briefly, ATCC 33591 was cultured in MHB overnight and treated with sub-inhibitory concentrations of BDMC for 4 h. The sample without BDMC was taken as a control. According to the manufacturer’s protocol, total RNA was prepared from *S. aureus* (ATCC 33591) using the E.Z.N.A.^®^ bacterial RNA kit (Omega Bio-tek, Norcross, GA, USA). The RNA concentration was evaluated by measuring the absorbance ratio at 260 nm on a spectrophotometer (BioTek, Winooski, VT, USA). Next, a QuantiTect reverse transcription kit (Qiagen, Dusseldorf, Germany) was used to synthesize the complementary DNA based on the manufacturer’s instructions. PCR was set up to a total volume of 20 μL as follows: 1 μL of each primer (10 μL/mL), 2 μL of sample cDNA, 10 μL of 2× SYBR Green PCR master mix (Life Technologies LTD, Warrington, UK), and 6 μL of deionized water. StepOne software v2.3 produced from Applied Biosystems (Waltham, MA, USA) was used to calculate the expression level of the target gene relative to the endogenous reference gene 16 rRNA through the delta–delta cycle threshold method.

### 4.5. ELISA

ELISA was performed using the previously described method [22]. Briefly, *S. aureus* ATCC 33591 in MHB was treated with sub-inhibitory concentrations of BDMC, and the untreated were taken as controls. The supernatants were collected after 4 h of incubation and 100 μL was added into a 96 well culture plate with 100 μL of RAW 264.7 cells. After incubating for 24 h, the mixture was collected and analyzed based on the mouse TNF-α or IL-6 ELISA MAX standard set according to the instructions.

### 4.6. Crystal Violet Biofilm Assay

The inhibition of the BDMC on the biofilm formation of *S. aureus* DPS-1, DPS-2, and ATCC 33591 was established in a crystal violet biofilm assay using a previously described protocol [36]. To each well of 96-well microtiter plates, 100 μL of overnight culture (0.5 MacFarland bacterial culture) was added and treated with sub-concentrations of BDMC. After 24 h at 37 °C, the planktonic cells were removed and washed with PBS three times, and each well of 96-well microtiter plates was stained with 1% (*w*/*v*) crystal violet for 10 min at room temperature, which was then rewashed for three times. The stained biofilms were solubilized in 100 μL of absolute ethanol, and the optical density (OD) values at 570 nm were measured. The percentage of biofilm inhibition was estimated using the formula below.

Percentage of inhibition = 100 − [(OD 570 nm of the treatment wells)/(OD 570 nm of the control wells) × 100)].

### 4.7. Statistical Analysis

The analyses were carried out in triplicate, and data were given as the mean ± standard deviation. The collected data were statistically analyzed using an independent Scheffe’s *t*-test (SPSS software version 22.0; IBM SPSS, Armonk, NY, USA). A statistically significant *p*-value of less than 0.05 was evaluated.

## Figures and Tables

**Figure 1 toxins-13-00804-f001:**
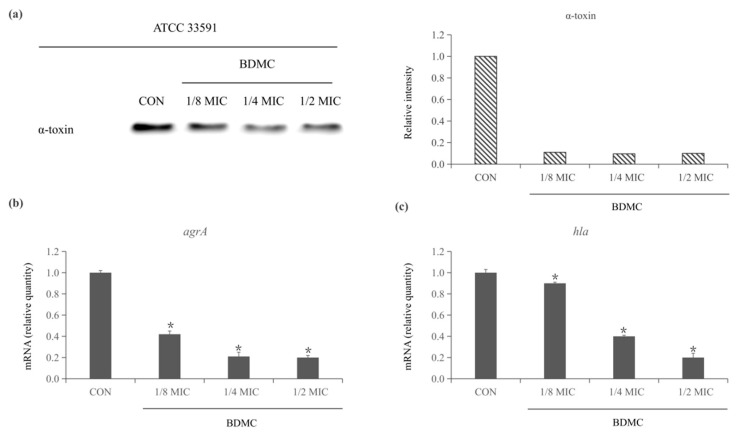
(**a**) Effect of BDMC treatment at sub-inhibitory concentrations on the expression of α-toxin in MRSA (ATCC 33591). CON was a control treated without BDMC. After treatment for 4 h, BDMC suppressed production of α-toxin significantly was determined through Western blotting. (**b**,**c**) The inhibitory effect of BDMC on the expression of enterotoxin gene *agrA* and *hla* was analyzed by qRT–PCR. After exposure to BDMC sub-inhibitory concentrations for 4 h, the expression of *agrA* and *hla* were down-regulated in a dose-dependent manner. The data were presented as the average of the three independent experiments with standard deviation. * Represents that *p* < 0.05.

**Figure 2 toxins-13-00804-f002:**
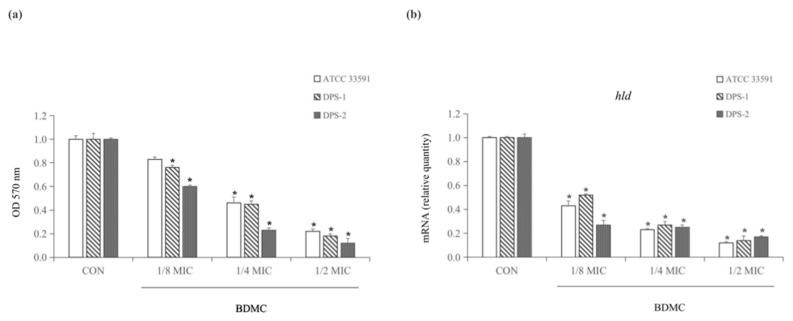
(**a**) Inhibitory effect of BDMC at sub-inhibitory concentrations on the biofilm formation of *S. aureus* standard strain (ATCC 33591) and clinical isolates DPS-1 and DPS-2. (**b**) In the presence of BDMC at sub-inhibitory concentrations, BDMC inhibited the expression of *hld* in MRSA (ATCC 33591, DPS-1, and DPS-2) cultures. The data were presented as the average of the three independent experiments with standard deviation. * Represents that *p* < 0.05.

**Figure 3 toxins-13-00804-f003:**
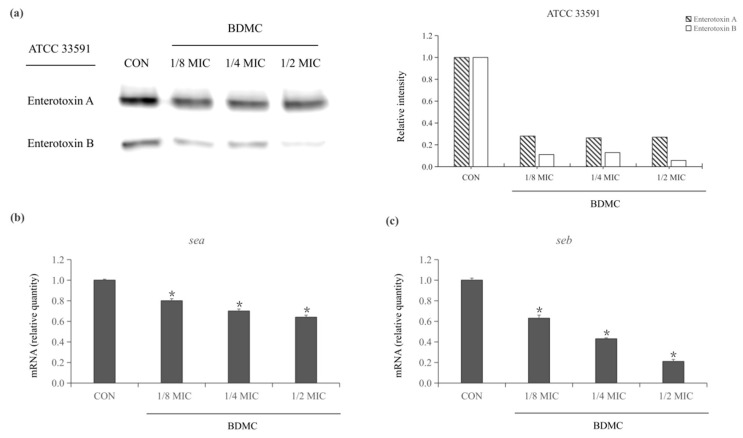
(**a**) BDMC at sub-inhibitory concentrations on enterotoxin A and enterotoxin B expression in MRSA (ATCC 33591). CON was a control treated without BDMC. (**b**,**c**) Relative expression of enterotoxin gene *sea* and *seb* in MRSA (ATCC 33591) cultures in the presence of sub-inhibitory concentrations of BDMC. Both enterotoxin genes were reduced in a dose-dependent manner. The data were presented as the average of the three independent experiments with standard deviation. * Represents that *p* < 0.05.

**Figure 4 toxins-13-00804-f004:**
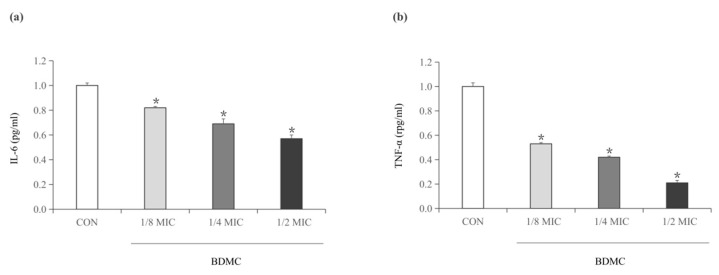
(**a**) IL-6 released from RAW 264.7 cells stimulated with *S. aureus* supernatants cultured in the presence of sub-inhibitory concentrations of BDMC. (**b**) TNF released from RAW 264.7 cells stimulated with *S. aureus* supernatants cultured in the presence of sub-inhibitory concentrations of BDMC. After stimulation for 24 h with RAW 264.7 cells, ELISA was used to measure the levels of IL-6 and TNF. The data were presented as the average of the three independent experiments with standard deviation. * Represents that *p* < 0.05.

**Table 1 toxins-13-00804-t001:** Primers used in in this study.

Primer	Sequence (5′-3′)
*16S*	F: ACTCCTACGGGAGGCAGCAG
	R: ATTACCGCGGCTGCTGG
*sea*	F: ATGGTGCTTATTATGGTTATC
	R: CGTTTCCAAAGGTACTGTATT
*seb*	F: TGTTCGGGTATTTGAAGATGG
	R: CGTTTCATAAGGCGAGTTGTT
*agrA*	F: TGATAATCCTTATGAGGTGCTT R: CACTGTGACTCGTAACGAAAA
*hla*	F: TTGGTGCAAATGTTTC R: TCACTTTCCAGCCTACT
*hld*	F: ATTTGTTCACTGTGTCGATAATCC R: GGAGTGATTTCAATGGCACAAG

## Data Availability

Personal information is included, so it was conducted for research only.

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
