# Peer review of "Bisdemethoxycurcumin Reduces Methicillin-Resistant Staphylococcus aureus Expression of Virulence-Related Exoproteins and Inhibits the Biofilm Formation"

_toxins, 2021, doi:10.3390/toxins13110804_

Round 1
Reviewer 1 Report
The manuscript deals with a topic of crucial interest in recent times, such as the research of new methods to treat microorganism-resistant strains of bacteria that are pathogenic for humans.
In addition, looking for natural substances derived from plants such as turmeric offers added value by not using chemical synthesis compounds that may have a greater risk of undesired effects.
The manuscript appears to be a continuation of a work already published where bisdemethoxycurcumin (BDMC) is presented as a potential growth inhibitor of the antibiotic resistant strain of Staphillococcus aureus. The present work considers the possibility that this compound is related to the inhibition of biofilm formation of the bacteria, which has already been described as one of the problems for its control and its generation of resistance to antibiotics; Furthermore, a brief study of the possible resistance that this strain may present to BDMC is presented. But none of these results is supported by more in-depth studies or in a model that gives a more concrete idea of ​​how it acts in the organism of some living being; Yes, it certainly does not generate resistance beyond the test presented by the manuscript that in 10 consecutive cultures it does not show that resistance is generated in culture media.
Therefore, I do not consider the data and what the manuscript consists of according to a publication in this journal and should be completed with more studies that will ensure the hypotheses raised and that can ensure a possible administration of BDMC as a potential inhibitor of the development of the illness.
Author Response
Dear Editors:
Thank you for your letter and the comments from the reviewers on our manuscript, "Bisdemethoxycurcumin Reduces Methicillin-resistant Staphylococcus aureus Expression of Virulence-related Exoproteins and Inhibit the Biofilm Formation" ( Manuscript ID: toxins-1441358 ).
Those comments are all extremely valuable and helpful in revising and improving our paper, as well as providing important guiding significance for our research. We have studied comments carefully and have made corrections which we hope to meet with approval.
As the editor suggested that, we have rewritten the sentences to reduce the repetition rate of the manuscript, and the grammar of the full text has also been checked and corrected.
The revised sections of the paper are marked in Track Changes. The following are the main corrections in the paper and responses to the reviewer's comments: (Reviewer 1, Reviewer 2, Reviewer 3)
1st, Nov. 2021
Author's Reply to the Review Report (Reviewer 1)
Dear Reviewers:
Thank you for your comments on our manuscript, "Bisdemethoxycurcumin Reduces Methicillin-resistant Staphylococcus aureus Expression of Virulence-related Exoproteins and Inhibit the Biofilm Formation". Those comments are all extremely valuable and helpful in revising and improving our paper, as well as providing important guiding significance for our research. We have studied comments carefully and have made corrections which we hope meet with approval. The revised sections of the paper are marked in Track Changes. The following are the main corrections in the paper and responses to the reviewer's comments:
Response to comment: None of these results is supported by more in-depth studies or in a model that gives a more concrete idea of how it acts in the organism of some living being.
Response: It is really true as Reviewer suggested that many of our studies are preliminary experiments. Therefore, we have revised the highlights of the article, and the current research results are more focused on providing foundational value. Such as ,and reference value was provided for the development of an antibiotic replacement, antibiotic adjuvant, or antibacterial lead compound. (There are supplementary explanations in the Abstract and Discussion section of the manuscript.)
Considering the questions raised by the reviewer, we added the qRT-PCR of biofilm-related genes(Figure 2 b)and deleted the drug resistance assay.
In addition, we will consider living being experiments in our future research.
Other changes:
We have revised the grammar of the entire manuscript, and replaced the high-resolution Figures. We tried our best to improve the manuscript and made some changes in the manuscript. These changes will not influence the content and framework of the paper. And here we did not list the changes but marked in Track Changes in revised paper. We appreciate for Reviewers’ warm work earnestly, and hope that the correction will meet with approval.
Once again, thank you very much for your comments and suggestions.

Reviewer 2 Report
Within this paper the Authors investigated the effects of bisdemethoxycurcumin (BDMC) on the production of virulence factors (exoproteins Enterotoxin A and Enterotoxin B, and α-toxin) by S. aureus, and on biofilm formation. This topic is relevant since anti-virulence compounds are taking hold as a potential approach to treat bacterial infection and drug resistance, and in this context the results presented appear very promising.
However, the work shows weakness, particularly regarding the lack of appropriate controls in western blot and RT-PCR experiments.
In Western blot experiments an antibody against a housekeeping protein must be used, to ensure that all the samples are homogenous, and that the decrease in intensity of the bands is actually due to a reduced production of the target proteins.
Similarly, in RT-PCR it is not clear if an internal control with a housekeeping gene has been performed. The description of the experimental procedure must be more accurate: PCR conditions are missing, as well as the equipment and the methods utilized for the quantification of the expressed genes.
The quality and the resolution of the figures should be improved. Finally, the language is quite clear and fluent, however the grammar needs some improvement.
Author Response
Dear Editors:
Thank you for your letter and the comments from the reviewers on our manuscript, "Bisdemethoxycurcumin Reduces Methicillin-resistant Staphylococcus aureus Expression of Virulence-related Exoproteins and Inhibit the Biofilm Formation" ( Manuscript ID: toxins-1441358 ).
Those comments are all extremely valuable and helpful in revising and improving our paper, as well as providing important guiding significance for our research. We have studied comments carefully and have made corrections which we hope to meet with approval.
As the editor suggested that, we have rewritten the sentences to reduce the repetition rate of the manuscript, and the grammar of the full text has also been checked and corrected.
The revised sections of the paper are marked in Track Changes. The following are the main corrections in the paper and responses to the reviewer's comments: (Reviewer 1, Reviewer 2, Reviewer 3)
1st, Nov. 2021
Author's Reply to the Review Report (Reviewer 2)
Dear Reviewers:
Thank you for your comments on our manuscript, "Bisdemethoxycurcumin Reduces Methicillin-resistant Staphylococcus aureus Expression of Virulence-related Exoproteins and Inhibit the Biofilm Formation". Those comments are all extremely valuable and helpful in revising and improving our paper, as well as providing important guiding significance for our research. We have studied comments carefully and have made corrections which we hope meet with approval. The revised sections of the paper are marked in Track Changes. The following are the main corrections in the paper and responses to the reviewer's comments:
Response to comment 1: In Western blot experiments an antibody against a housekeeping protein must be used.
Response: It is really true as Reviewer suggested that an antibody against a housekeeping protein must be used. The current report used GAPDH as a housekeeping protein to evaluate the reduction in target protein production. Because the literature we referred to did not show the bands of GAPDH in the results, we did not show GAPDH in the manuscript either. We are very sorry about this. In future experiments, we will pay great attention to this problem.
Response to comment 2: In qRT-PCR it is not clear if an internal control with a housekeeping gene has been performed. The description of the experimental procedure must be more accurate: PCR conditions are missing, as well as the equipment and the methods utilized for the quantification of the expressed genes.
Response : We are very sorry that we did not clearly describe the method. Considering the reviewer’s suggestion, we supplement an explanation of the method. (Line-475-484, A StepOne software v2.3 produced from Applied Biosystems was used to calculate the expression level of the target gene relative to the en-dogenous reference gene 16 rRNA through the delta-delta cycle threshold method.)
Response to comment 3: The quality and the resolution of the figures should be improved.
Response: We increased the resolution of all Figures from 300dpi to 600dpi.
Other changes:
We have supplemented the summary and discussion. We added the qRT-PCR of biofilm-related genes(Figure 2 b)and deleted the drug resistance assay.
We have revised the grammar of the entire manuscript, and we tried our best to improve the manuscript and made some changes in the manuscript. These changes will not influence the content and framework of the paper. And here we did not list the changes but marked in Track Changes in revised paper. We appreciate for Reviewers’ warm work earnestly, and hope that the correction will meet with approval.

Reviewer 3 Report
This manuscript describes inhibitory effect of toxin production (hemolysin A, SEA, SEB) and biofilm formation of MRSA by sub-MIC concentration of BDMC. Experiments appear to be performed appropriately and results are well presented. This reviewer points out following comments to improve this manuscript.
1. Throughout the manuscript, usage of technical terms should be checked and corrected.
1)As shown in line 40, α-toxin means hemolysin A (alpha-hemolysin) encoded by hla. In such case, authors should use only one term of toxin to keep uniformity in the manuscript. Both α-toxin and alpha-toxin were used.
2)Both bisdemethoxycurcumin and BDMC were used in text. Once BDMC was defined, use it throughout the manuscript. (see line 194, 195, etc.)
3)Line 160: Use “MRSA” one abbreviation was defined. Check other part of manuscript.
2. This reviewer does not understand “Drug resistance study” shown in Results as well as Methods section. What is the meaning/significance of this study? This author imagines that resistance to BDMC did not occur after 10 times serial culture. If authors want to demonstrate it, more different strains and more serial cultures should be done. This experiment seems to be too preliminary, and thus may be deleted. Authors may mention it in the text (Discussion) that preliminary study suggested no occurrence of resistance to BDMC, but further investigation would be necessary
Author Response

(The authors gave the same response as above.)

Round 2
Reviewer 1 Report
As I commented in the fisrt version, the subject of this manuscript is very interesting nowadays.
I have verified that the manuscript has been given a good review, some of the things that were weaker have been clarified and the figures and discussion part has been improved.
The work still seems to me that it contains improvable parts, but I value the work done in the review and the authors' responses to consider its publication in the journal as feasible.
Author Response
Reply to Academic Editor
Dear Academic Editor:
Thank you very much for giving us an opportunity to revise our manuscript, I appreciate the editor and reviewers very much for their positive and constructive comments and suggestions on our manuscript.
Thanks to the editor for the optimization of the grammar of the manuscript and the checking of the format. Considering the editor and Reviewer’s suggestion, we made minor revisions to the manuscript and marked in Track Changes in the revised paper.
Response to comment: (Figure 4) The same results twice? (Line 147)
Response: The result is the same, the figure is replaced with a higher resolution picture.
We would like to express our great appreciation to you for your comments on our paper.
Thank you and best regards.
11th, Nov. 2021
---------------------------------------------------------------------------------------------------------------------------------
Author's Reply to the Review Report (Reviewer 1)
Dear Reviewer:
Thank you very much for giving us an opportunity to revise our manuscript, we appreciate you very much for your positive and constructive comments and suggestions on our manuscript
The manuscript has been optimized for grammar and spells checking and marked in Track Changes in the revised paper.
We would like to express our great appreciation to you for your comments on our paper.
Thank you and best regards.
11th, Nov. 2021

Reviewer 2 Report
With this revised version, the manuscript has been sufficiently implemented, and in my opinion it can be accepted for publication.
Author Response
Reply to Academic Editor
Dear Academic Editor:
Thank you very much for giving us an opportunity to revise our manuscript, I appreciate the editor and reviewers very much for their positive and constructive comments and suggestions on our manuscript.
Thanks to the editor for the optimization of the grammar of the manuscript and the checking of the format. Considering the editor and Reviewer’s suggestion, we made minor revisions to the manuscript and marked in Track Changes in the revised paper.
Response to comment: (Figure 4) The same results twice? (Line 147)
Response: The result is the same, the figure is replaced with a higher resolution picture.
We would like to express our great appreciation to you for your comments on our paper.
Thank you and best regards.
11th, Nov. 2021
Author's Reply to the Review Report (Reviewer 2)
Dear Reviewer:
Thank you very much for giving us an opportunity to revise our manuscript, we appreciate you very much for your positive and constructive comments and suggestions on our manuscript.
We have made corrections according to your comments, the manuscript has been optimized for grammar and spells checking and marked in Track Changes in the revised paper.
We would like to express our great appreciation to you for your comments on our paper.
Thank you and best regards.
11th, Nov. 2021
